# Novel Protein-Based Pneumococcal Vaccines: Assessing the Use of Distinct Protein Fragments Instead of Full-Length Proteins as Vaccine Antigens

**DOI:** 10.3390/vaccines7010009

**Published:** 2019-01-19

**Authors:** Theano Lagousi, Paraskevi Basdeki, John Routsias, Vana Spoulou

**Affiliations:** 1First Department of Paediatrics, “Aghia Sophia” Children’s Hospital, Immunobiology Research Laboratory and Infectious Diseases Department “MAKKA,” Athens Medical School, 11527 Athens, Greece; basdeki.evi@gmail.com (P.B.); vspoulou@med.uoa.gr (V.S.); 2Department of Microbiology, Athens Medical School, 11527 Athens, Greece; jroutsias@med.uoa.gr

**Keywords:** protein-based pneumococcal vaccines, peptide antigens, vaccine adjuvants

## Abstract

Non-serotype-specific protein-based pneumococcal vaccines have received extensive research focus due to the limitations of polysaccharide-based vaccines. Pneumococcal proteins (PnPs), universally expressed among serotypes, may induce broader immune responses, stimulating humoral and cellular immunity, while being easier to manufacture and less expensive. Such an approach has raised issues mainly associated with sequence/level of expression variability, chemical instability, as well as possible undesirable reactogenicity and autoimmune properties. A step forward employs the identification of highly-conserved antigenic regions within PnPs with the potential to retain the benefits of protein antigens. Besides, their low-cost and stable construction facilitates the combination of several antigenic regions or peptides that may impair different stages of pneumococcal disease offering even wider serotype coverage and more efficient protection. This review discusses the up-to-date progress on PnPs that are currently under clinical evaluation and the challenges for their licensure. Focus is given on the progress on the identification of antigenic regions/peptides within PnPs and their evaluation as vaccine candidates, accessing their potential to overcome the issues associated with full-length protein antigens. Particular mention is given of the use of newer delivery system technologies including conjugation to Toll-like receptors (TLRs) and reformulation into nanoparticles to enhance the poor immunogenicity of such antigens.

## 1. Introduction

*Streptococcus pneumoniae* is a major cause of morbidity and mortality worldwide and antibiotic treatment is being compromised by the emergence of multidrug-resistant strains [1,2,3]. To date, 95 distinct capsular serotypes have been identified including the recently reported serotypes 6D, 11E, 20A/B, and 35D [4,5].

Two types of pneumococcal vaccines both targeting capsular polysaccharides are currently licensed: (a) the 23-valent pneumococcal polysaccharide-based vaccine (PPV-23) and (b) the 7-, 10-, and 13-valent pneumococcal conjugate vaccines (PCV-7, -10, -13). PPV-23 is poorly immunogenic in children under two years of age and generates neither an immune memory response nor herd immunity [6]. Conjugation of a pneumococcal polysaccharide to a carrier protein turned polysaccharide-based vaccines from T-cell independent to T-cell dependent antigens, enhancing their immunogenicity. Conjugate vaccines evoked an efficient immune memory and induced relevant herd immunity through a significant effect on nasopharyngeal carriage [7].

All currently available pneumococcal vaccines have limitations. Both PPV and PCVs are serotype-based vaccines and therefore they elicit only serotype-specific immunity. Emergence of replacement serotypes has repeatedly occurred after their introduction, even after the use of expanded valency PCVs [8,9,10,11], not to mention the vaccine-failures reported following the immunization with PCV-13 [12,13]. PCVs are less efficient against non-invasive pneumococcal disease (IPD) in comparison to IPD [14,15,16,17,18]. Their serotype coverage in developed countries is substantially higher compared to the developing world where pneumococcal disease is caused by a wider spectrum of serotypes [19]. Their debatable effect on pneumococcal disease has been well documented among high-risk individuals [20]. PCVs are very complicated and expensive to manufacture and thus unaffordable for those in greater need, especially in developing countries. Notably, PCV-13 is the second most expensive vaccine for the pediatric population among the 31 listed in Centres for Diseases Control and Prevention (CDC) (https://www.cdc.gov/vaccines/programs/vfc/awardees/vaccine-management/price-list/index.html#f5) [21].

An initial attempt to overcome the replacement phenomenon and the great geographical variation in circulating serotypes was the addition of new pneumococcal serotypes in the already licensed PCVs. PCV-15 is such a vaccine, containing the PCV-13 serotypes as well as 22F and 33F. It is currently on phase 3 clinical trial [22,23]. However, such vaccines will most likely retain all the shortcomings of PCVs mentioned above.

Another approach employs the use of next-generation whole-cell (killed) pneumococcal vaccines (WCPVs). So far, several methods including attenuation, chemical treatment, or preparations of whole-cell crude extracts have been used to reduce or inactivate a pathogen’s virulence for the development of vaccines. For example, both whole-cell and acellular purified protein-based pertussis vaccines provide efficient protection; however, whole-cell pertussis vaccines are more efficacious and produce longer lasting immunity than acellular vaccine. Based on such observations, WCPVs may be safe, provide multiple common surface proteins in native configurations and induce an immune response that is protective across different serotypes [24].

A WCPV may be the most suitable strategy for developing countries, as it is inexpensive and easy to construct and administer [25].

One such vaccine containing a non-capsulated pneumococcal strain has shown protection against nasopharyngeal (NP) colonization and sepsis in immunized mice [26]. Protection against NP colonization was shown to be CD4+ Interleukin 17A (IL-17A)-dependent and non-capsular antibody mediated [26,27].

The WCPV was used in two clinical trials to study safety, dose tolerance, and immunogenicity [28]. In healthy US adults (NCT01537185) WCPV was shown to be safe and well tolerated; significant IgG responses to pneumococcal antigens were elicited, including pneumococcal surface protein A (PspA) and pneumolysin, and functional antibody responses were detected in a pneumolysin toxin neutralizing assay [29]. Significant increases in T-cell cytokine responses, including IL-17A, were measured among subjects receiving the highest dose of WCPV. Passive transfer of human immune sera protected mice against fatal sepsis when challenged intravenously with a virulent strain of pneumococci [30]. Further studies are required to evaluate the safety, dose tolerance, and immunogenicity of such vaccines.

## 2. Protein-Based Pneumococcal Vaccines

### 2.1. Rationale towards Protein-Based Pneumococcal Vaccines

A promising alternative to capsular polysaccharide-based vaccines is the use of pneumococcal virulence proteins for the construction of novel vaccines. Pneumococcus has several proteins that are well-conserved among different serotypes, most of which are surface-displayed and thus antibody accessible [31]. Similar to capsular polysaccharide antigens, non-capsular protein antigens may induce the production of opsonophagocytic antibodies, with the potential to offer serotype-independent protection against pneumococcal disease [32]. Considering the fact that pneumococcal proteins contribute to bacterial virulence, as they are involved in several different stages of disease pathogenesis, anti-protein antibodies, unlike anti-capsular antibodies, may display additional protective mechanisms other than opsonophagocytosis. For instance, human antibodies against several pneumococcal proteins have been shown to reduce pneumococcal adherence to human lung epithelial cells and murine NP colonization by pneumococcus [33].

Soon, several epidemiological and experimental findings suggested that not only humoral but also cellular immunity generated by pneumococcal proteins is involved in the natural mechanism of acquired immunity to pneumococcal disease [34]. Prior to the PCV introduction, age-related decreases in IPD rates preceded the development of measurable serotype-specific antibodies [35]. IPD incidence peaks at around 10 months of age and then drops by 50% till the age of two years almost equally among all serotypes, well before the development of protective levels of anti-polysaccharide antibodies, arguing in favor of a serotype-independent mechanism [35]. This could be explained by the acquisition of antibodies to non-capsular conserved antigens, following repeated mucosal exposure to pneumococci, albeit such an assumption has not been confirmed so far. Indeed, the increased IPD susceptibility of not only pediatric, but also AIDS adults (i.e., HIV-infected adults), with low CD4+ T-cells implies a protective role of CD4 T-cells independent of antibodies [36,37,38]. Therefore, proteins as vaccine antigens are likely to induce both arms of immunity.

Many pneumococcal proteins have been undergoing preclinical evaluation as possible vaccine candidates [31]; Table 1 summarizes pneumococcal proteins that have been evaluated or are currently under evaluation as components of protein-subunit vaccines in humans. The majority of such formulations combines several proteins in order to expand their coverage and minimize the possibility of immune escape [39,40].

### 2.2. Challenges Concerning the Licensure of Protein-Based Pneumococcal Vaccines

Protein-based pneumococcal vaccines face several challenges that remain to be addressed before licensure. The amino-acid sequence and level of expression vary among different serotypes [51]. The amino-acid sequence among pneumococcal proteins shows much less diversity compared to the biochemical structure among pneumococcal polysaccharides. However, some proteins contain regions with high sequence variability among different serotypes, which make them non-suitable candidates for the development of a serotype-independent vaccine. The level of protein expression among different strains/serotypes displays significant heterogeneity and variability. A recent study of the pneumococcal variome analyzed the distribution of proteins associated with virulence and host–pathogen interactions among 25 pneumococcal strains and revealed that only 26 of the 65 proteins are constitutively expressed without significant sequence variation in all strains [52]. Remarkably, only three of the above 26 widely expressed proteins (namely pneumococcal surface antigen A (PsaA), serine threonine kinase protein (StkP) and pneumolysin), have been so far included as candidate antigens in the vaccines that are currently under evaluation in clinical studies listed in Table 1. Sequence and expression variability could be overcome by combining more than one protein antigens in one molecule; though such an approach may raise difficulties concerning its high-cost and stable production. It is usually the protein’s 3-dimensional (3-D) structure that is immunogenic. The synthesis of recombinant protein antigens is characterized by a high degree of chemical instability that hampers their synthesis for mass production. Recombinant proteins, when overexpressed in a heterologous host, are often misfolded, exposing their hydrophobic regions and forming insoluble aggregates blocking their proper production [53]. Various cryptotopes may be exposed and immunodominant epitopes may be buried due to the denaturation procedure, preventing a more specific immunological response [54]. Pneumococcal proteins (i.e., pneumolycin) may be associated with undesirable reactogenicity or even toxicity if used in humans while the cost required for their detoxification is quite high [55,56]. Using whole proteins as vaccine antigens cannot exclude the possibility of a host autoimmune response caused by certain antigen fragments. For instance, a prior report suggested that some antibodies to the protection-eliciting coiled coil domain of PspA may react with denatured myosin at room temperature [55,56]. Although there is no compelling evidence for the clinical relevance of this observation, further research is required for the development of a PspA-based vaccine containing this domain [56,57].

Another issue worth mentioning is the different effects of polysaccharide-based and protein-based pneumococcal vaccines on NP colonization. NP colonization is an essential precursor to pneumococcal disease pathogenesis. In the NP, the expression of various pneumococcal virulence factors differs between transparent and opaque phase, facilitating the establishment of commensal carriage. Therefore, antibodies against capsular polysaccharides fail to protect against already established NP colonization, while antibodies against proteins may hamper initiation of colonization. Besides, antibodies against capsular polysaccharides prevent carriage of a specific serotype leading to a significant herd immunity effect; however, complete elimination of carriage in the NP and subsequent replacement of vaccine serotypes by non-vaccine serotypes has been an important issue. In contrast, protein-based pneumococcal vaccines, that are serotype independent, have been associated with intermittent priming and boosting to all pneumococcal serotypes through natural exposure [58]. This effect has been suggested to retain pneumococcal carriage density below disease threshold, instead of the total elimination induced by polysaccharide-based vaccines. Thus, such vaccines may induce more robust memory responses, and at the same time, prevent other pathogens emerging in the nasopharynx. A recent study among healthy infants showed that NP mucosal antibody levels to pneumococcal histidine triad D (PhtD), pneumococcal-choline binding protein A (PcpA), and pneumolysin detoxified derivative (PlyD1), induced through repeated colonization, correlated with protection against acute otitis media (AOM) but not protection against NP colonization [59]. Additionally, in a mouse influenzae/pneumococcal co-infection model it was shown that vaccination with PhtD prevented pneumococcal density from reaching a pathogenic threshold during a viral upper tract respiratory infection (URTI) without elimination of the pathogen from the NP [60]. However, things may differ in developing countries, where persistently high antigenic exposure attributed to the high pneumococcal burden might lead to immune hyporesponsiveness, resulting in low vaccine efficacy, since antigen specific CD4 T-cells can lose their ability to help B cells differentiate into antibody secreting cells [61].

AOM is the most suitable type of pneumococcal infection to assess the efficacy of pneumococcal vaccines due to its easy diagnosis using tympanocentesis and lower costs and shorter time to accumulate a sufficient number of subjects compared to IPD with concurrent PCV administration. Mucosal immunity plays a critical role in control of pneumococcus locally invasive infections including AOM. Capsular polysaccharide vaccines are less effective in protection against AOM than against IPD. As mentioned above, NP mucosal antibody levels to PhtD, PcpA, and PlyD1, induced through repeated colonization, correlated with protection against AOM [59]. Similarly, higher levels of mucosal antibodies against *6* pneumococcal protein antigens (PhtD, Pneumococcal histidine triad protein E (PhtE), the endo-β-*N*-acetylglucosaminidase (LytB), PcpA, PspA, and Ply) induced through natural exposure in healthy children were associated with reduced risk of AOM [33,62,63].

Currently, a clinical efficacy study is underway in an otitis-prone Native American population (NCT01545375). Nevertheless, for a vaccine to be efficient against AOM, special attention should be given to studies in the infection-prone child who has an immune response that is generally deficient compared to non-otitis prone children leading to susceptibility to recurrent AOM.

Therefore, while otitis-prone population may be attractive to assess the efficacy of protein-based pneumococcal vaccines against AOM, it should be realized that the response to such vaccines in otitis-prone populations may not be of the same magnitude as the general population.

## 3. Distinct Antigenic Fragments within Pneumococcal Proteins as Vaccine Antigens

### 3.1. Rationale towards the Use of Distinct Protein Fragments as Vaccine Antigens

To overcome the shortcomings of the protein-based pneumococcal vaccines mentioned above, extensive research has been directed towards the identification of highly-conserved antigenic regions, peptides, or even distinct B-cell epitopes within pneumococcal proteins that could serve for the design of novel pneumococcal vaccines. This approach is based on the theory that in order to initiate an efficient immune response against a specific protein, a vaccine principally needs to include only the minimal immunogenic peptide sequence which can be produced synthetically. These epitopes correspond to only a small region of the entire protein (polypeptide) antigen. Vaccines containing distinct antigenic protein fragments are capable of inducing immune responses through the T-cell-dependent pathway requiring both B-cell and T-cell peptide-epitopes specifically recognized by the T-cell and B-cell receptor (TCR and BCR respectively). Currently, several human peptide vaccines are in different stages of clinical testing while the only peptide vaccines in Phase III of clinical studies are cancer therapeutic vaccines [64].

### 3.2. Discovery of Antigenic Regions within Pneumococcal Proteins

Several methods have been used so far for the identification of antigenic regions within pneumococcal proteins. Among the standardized procedures, the most common one uses a panel of monoclonal antibodies to different domains and epitopes and independently evaluates their in vitro opsonophagocytic ability and in vivo protective efficacy. A more recent approach [65] uses human sera from individuals with natural infection as primary screening. Such an approach offers the possibility of identifying proteins that are not predicted by Open reading frame (ORF)-finding algorithms and allows a rapid selection of in vivo expressed antigens [66]. Following this, selected antigens are evaluated in preclinical studies to assess their potential as vaccine candidates. Based on this method, Giefing et al. identified large antigenic regions within several pneumococcal proteins (of a total length of 13494 amino acids), consistently recognized by adult patients with IPD [67]. Similarly, Beghetto et al. identified a panel of antigenic fragments within surface pneumococcal proteins; namely, the choline-binding protein D (CbpD), the pneumococcal histidine triad proteins PhtD and PhtE, the PspA, the plasminogen and fibronectin binding protein B (PfbB), and the zinc metalloproteinase B (ZmpB) [68]. However, in both reports mentioned above, the long length of the identified antigenic fragments hampers the construction of synthetic analogues that could be used as stable vaccine components. Therefore, a more recent study proceeded the identification of immunodominant B-cell epitopes within the antigenic regions previously identified by Beghetto et al. using a pediatric patient cohort with IPD [69]. The use of peptides in the design of pneumococcal vaccines is experiencing new enthusiasm due to the recent developments in peptide selection using different tools such as structural analysis by X-ray crystallography, Nuclear magnetic resonance (NMR) spectroscopy, mass spectrometry, and high-throughput screening techniques such as combinatorial chemistry, proteomics, genomics, and bioinformatics [70,71]. However, the selected peptides, regardless of the method used, require further evaluation of their immunogenicity and protective efficacy to confirm their potential as alternative vaccine candidates. Table 2 shows the most studied proteins regarding the identification of the antigenic regions they contain.

### 3.3. Potential Advantages of Antigenic Regions/Peptides within Pneumococcal Proteins Compared to Full-Length Proteins as Vaccine Candidates

Peptide-based vaccines offer several benefits compared to whole protein-based vaccines, including chemical stability and easier production processes, as they do not require folding into a tertiary 3-D structure [72]. Conjugation of various peptides from different antigens to the same carrier is much easier than conjugation of recombinant proteins. Such vaccines are considered safer than protein-based vaccines as they allow the exclusion of antigen fragments associated with autoimmune responses or toxicity. Synthetic peptides are inexpensive, requiring a much cheaper manufacturing process than recombinant proteins.

Synthetic peptides may induce better T-cell responses as vaccine antigens compared to full protein vaccines [73,74]. In fact, peptides are more efficiently endocytosed, processed, and presented on major histocompatibility complex (MHC) molecules compared to full-length proteins. Another, less investigated aspect of the antigen handling by antigen presenting cells (APCs), indicates that antigen cross-presentation—on a mole-for-mole ratio—is better for peptides than proteins. This is likely attributed to the efficient translocation of peptides into the cytoplasm from endosomes [75]. Sophisticated delivery systems can be easily applied in the design of peptide-based pneumococcal vaccines [72]. Their synthetic chemical stability allows an easier development of standardized in vitro assays in order to measure correlates of protection. Such immunological correlates may not be limited in the induction of opsonophagocytic antibodies, but also in the impairment of bacterial adherence to human nasopharyngeal and/or lung epithelial cells or in the induction of an efficient T-helper 17-mediated immune response to modulate neutrophil recruitment in the nasopharynx. A more comprehensive understanding of various factors influencing vaccine responses in a particular population/geography would be more easily defined using small synthetic peptides for vaccine development and deployment strategies instead of full-length recombinant proteins. Thus, the issue of constant natural exposure in developing countries associated with lower vaccine efficacy of protein-based pneumococcal vaccines would most likely be easier to overcome using peptides as vaccine antigens.

### 3.4. Protein Fragments as Vaccine Antigens

Significant progress has been achieved so far on the identification of antigenic regions, peptides, and/or B-cell epitopes within pneumococcal proteins and their evaluation as pneumococcal vaccine candidates.

PspA is a widely studied pneumococcal protein, expressed on the surface of all isolated strains, interfering with complement and C-reactive protein (CRP) deposition on the bacterial surface and immune adherence to erythrocytes [76,77,78,79]. Besides, PspA binding to lactoferrin via electrostatic interactions, specifically blocks this bactericidal peptide, preventing it from penetrating the bacterial membrane and killing pneumococcus [80]. PspA contains a variable α-helical N-terminal domain, an antigenically conserved proline-rich region (PRR), which is often interrupted by a non-proline block (NPB) and a choline-binding domain that anchors the protein to the bacterial outer cell wall [81,82]. Due to the high degree of amino-acid sequence variability among different pneumococcal strains, several PspA fragments mainly containing the N-terminal region, have been studied as vaccine candidates so far. Immunization of healthy adults with a single recombinant fragment of PspA containing the N-terminal region in a phase I clinical trial induced cross-reactive antibodies [49] that were able to induce passive protection in mice [47]. PspA fragments corresponding to the complete N-terminal regions of PspA protected mice against IPD caused by a strain expressing homologous PspA [83]. Remarkably, immunization of mice with a distinct 100-amino-acid fragment located within the more conserved N-terminal region of PspA induced protection in mice against pneumococcal lethal challenge [84,85]. In agreement with these reports, Roche et al. demonstrated the importance of C-terminal 104 and N-terminal 115 amino acids of the alpha-helical region of PspA in cross-protection of mice against IPD [83]. Considering that pneumococcal colonization is an essential precursor for disease pathogenesis, a more recent study reported that mice vaccination with recombinant salmonella outer membrane vehicles displaying the N-terminal part of the coiled-coil domain of PspA strongly reduces pneumococcal density in the nasopharynx, through an IL-17A dependent pathway [86]. The presence of one distinct epitope, common in different clinical isolates, strongly correlated with the induction of nasal IL-17A, that was, in turn, associated with a reduction of pneumococcal colonization [87].

Except from the N-terminal domains, immunization with recombinant PR (rPR) molecules and passive immunization with monoclonal antibodies reactive with either NPB or PR epitopes, alone or in combination with N-terminal fragments, were shown to be protective against IPD in mice [81,88,89]. However, there are concerns about PspA PR potential to cross react with the human protein myosin, which could induce autoimmune cardiac disorders [56]. Such an assumption has not been confirmed by Perciani et al., who reported that the presence of cross-reactive antibodies does not represent the risk of the development of an autoimmune response. A more recent study demonstrated that synthetic virus-like particles (SVLPs) carrying distinct B-cell epitope mimetics of length of no larger than 30-amino acids derived from the PR of PspA are highly immunogenic in mice without additional adjuvants and elicit cross-reactive significantly protective antibodies [90].

Histidine triad (Pht) protein family includes four highly conserved surface-exposed proteins (PhtA, PhtB, PhtD, and PhtE), characterized by the presence of five to six histidine triad (HxxHxH) motifs [91,92]. Pht proteins inhibit surface complement deposition and mediate bacterial adherence through zinc binding [91,92,93]. PhtD has been shown to display the least variability and is expressed by all pneumococcal strains screened thus far, with a 97–100% identity across different strains. Pht proteins have been widely considered as promising vaccine candidates. Indeed, immunization of mice with truncated derivatives from the C-terminal half of PhtD significantly increased their survival time after challenge with pneumococcus compared to immunization with the whole protein [94,95]. Further investigation revealed that immunization of mice with distinct B-cell epitopes within PhtD and PhtE proteins enhanced the survival of immunized mice compared to unimmunized controls when challenged with the high virulence pneumococcal serotype 3 [96]. Similar results were obtained for two distinct B-cell epitopes within CbpD involved in adherence, complement inhibition, and competence-induced cell lysis [97] and the bacterial adhesin ZmpB [98].

Finally, distinct antigenic regions have been identified within PsaA, mainly involved in manganese and zinc transportation, and a novel surface antigen encoded by ORF *spr1875* in the R6 strain genome, involved in pneumococcal sepsis. In details, it has been reported that 15-mer peptides within PsaA protein selected through biopanning of a phage display library are immunogenic and protective in mice [99], while a 161 amino acid-long fragment within Spr1875 conferred immunoprotection in mice against experimental sepsis [100].

### 3.5. The Combination of Whole Proteins with Protein Fragments Demonstrated Protective Efficacy against Multiple Diseases in Mice

Another approach combines full-length proteins with distinct protein fragments in order to enhance their potential as vaccine antigens. Mice immunized with PsaA-PspA fragments (containing both family1 and 2 N-terminal region clade-defining region) were protected against fatal challenge with pneumococcal strains expressing different PspAs regardless of the challenge route [101]. Similarly, peptides derived from choline-binding protein A (CbpA), involved in adherence to cytokine-activated lung cells and complement deposition, genetically fused to detoxified pneumolycin, a cholesterol-dependent cytolysin that induces pore formation in the membrane of eukaryotic cells, protected mice from pneumococcal carriage, otitis media, pneumonia, bacteremia, meningitis, and meningococcal sepsis in active as well as passive immunization models [102]. Goulart et al. showed that PspA fragments including N-terminal domains plus PR regions fused to pneumolycin-derivatives increase the immune response mediated by cross-reactive antibodies and complement deposition to heterologous strains, confer protection against fatal challenge and reduce nasopharyngeal colonization in mice [103,104]. Chen et al. showed that multivalent pneumococcal protein vaccines comprising pneumolysoid with epitopes/fragments of CbpA and/or PspA elicit strong and broad protection in sepsis, meningitis, and focal pneumonia animal models [105].

### 3.6. Particular Adjuvants and Delivery Methods

Protein fragments as vaccine antigens, even those comprising optimal B-cell and T-cell epitopes, are poorly immunogenic. Suboptimal priming can be potentially circumvented using carriers and adjuvants that allow the targeting of APCs and contribute to an adequate immune stimulation. The most extensively used adjuvant so far is alum. Alum induces a TLR-dependent effect, increases antigen stability and enhances delivery to APCs. Although alum has been licensed for human vaccines for decades and is still widely used, its use in humans meets several obstacles. Alum does not have the ability to elicit T-helper 1 type immunity or cytotoxic T-cell responses and vaccines containing alum adjuvant cannot be sterilized by filtration, frozen, or lyophilized [106]. Another widely used immunostimulator in preclinical studies is Freud’s adjuvant. However, the highly effective Freund’s adjuvant is also reactogenic and frequently induces granulomas, sterile abscesses, and ulcerative necrosis at the site of inoculation, which precludes its use in human vaccines [107]. Another approach includes direct conjugation of long peptides to TLRs that may result in improved T-cell activation and antigen presentation leading to a more efficient vaccine. The presence of the antigen and the TLR ligand in the same endosomes determines the entrance to the presentation pathways [108]. Conjugation of antigenic peptides to TLR ligands like the TLR9-ligand CpG or the TLR1/2 heterodimer agonist Pam3CSK4 has been shown to strongly improve T-cell priming in vivo due to the combined effect of an increased uptake of long peptides and co-delivery with the immune-stimulatory signal [109,110]. Furthermore, the Pam3CSK4-conjugates were able to establish potent anti-tumor immune responses in multiple preclinical models and are now being tested in a phase I/II clinical trial evaluating synthetic peptide vaccination for treatment of Human Papillomavirus (HPV)-induced cancers [111,112]. Pneumococcal proteins when co-administrated with TLRs have been shown to induce significantly enhanced immunogenicity and protective efficacy in different animal models [113,114]. To the same direction, TLRs represent a promising delivery platform for the design of neo-epitope-based pneumococcal peptide vaccines. Furthermore, various advanced delivery systems have been studied so far to achieve the design of more efficient peptide-vaccines [115]. Over the past decade, nanoscale size (<1000 nm) materials such as virus-like particles (VLPs), outer-membrane vehicles (OMVs), liposomes, immune stimulating complexes (ISCOMs), polymeric, and non-degradable nanospheres have received attention as potential delivery vehicles for vaccine antigens which can both stabilize vaccine antigens and act as adjuvants. Besides, they offer the ability to design vaccines containing multiple protein antigenic fragments that specifically target immune cells, leading to more effective uptake by APCs [116,117,118,119,120]. VLPs, previously used as delivery system for pneumococcal protein antigens, are made from synthetic coiled-coil lipopeptides able to display multivalent epitope mimetics enhancing their presentation to the immune system, without the need for external adjuvants. VLPs are considered versatile tools in vaccine development due to their favorable immunological characteristics such as their size, repetitive surface geometry, their ability to induce both innate and adaptive immune responses, safety, and low cost. Several VLP-based vaccines are currently undergoing preclinical and clinical evaluation, [90,120] while VLP-based vaccines against HPV such as Cervarix^®^, Gardasil^®^, and Gardasil9^®^ and Hepatitis B Virus (HBV) including the 3rd generation Sci-B-Vac™ are available on the market [121,122]. In addition, the first licensed malaria-VLP-based vaccine Mosquirix™ has been recently approved by the European regulators [123]. Bacterial OMVs represent another promising delivery system among nanoparticles used so far as delivery system for pneumococcal protein antigens. OMVs are vesicles of lipids released from the outer membranes of gram-negative bacteria to communicate among themselves and with other microorganisms in their environment. OMVs, displaying essential immunostimulatory properties, with the ability of antigen presentation on their surface and incorporation of heterologous antigens may be used as potent vaccine delivery systems. Such OMVs have been used for the design of the meningococcal B vaccine (Bexsero, Novartis) [124]. As for pneumococcus, PspA fragments and full-length proteins incorporated in Salmonella OMVs induced strong protection in a murine model of pneumococcal colonization, when intranasally administered, without the need for a mucosal adjuvant, in T-helper 17-mediated way [86,87].

Particular vaccine adjuvants or delivery systems used in peptide-based pneumococcal vaccines in pre-clinical studies so far are shown in Table 3.

## 4. Conclusions

Current PCVs have been highly successful in reducing the incidence of IPD due to vaccine serotypes. However, several issues mainly associated with the emergence of non-vaccine serotypes have been a consistent, recurring problem.

Pneumococcal proteins that are universally expressed among all serotypes have been considered an interesting alternative for the development of serotype-independent pneumococcal vaccines. However, such an approach is reaching a crossroads regarding further evaluation.

A promising alternative may employ the use of antigenic protein fragments and peptides, several of which have been evaluated as vaccine antigens in pre-clinical studies, alone or in combination with whole pneumococcal proteins, as vaccine candidates. Such antigens retain the benefits of protein antigens i.e., immunogenicity in infants, induction of immune memory and natural serotype-independent “priming/boosting”. In the case of peptides, their low-cost and stable construction permits the combination of several highly-conserved antigens with the potential to activate both cellular and humoral immune responses and impair different stages of pneumococcal disease, offering even wider serotype coverage and more efficient protection. These vaccines would take advantage of newer delivery system technologies including conjugation to TLRs and reformulation of peptide antigens into nanoparticles that significantly enhance their immunogenicity. Such approaches would further benefit from the great advances in X-ray crystallography and NMR spectroscopy that allow precise structure determination, leading to synthesis of peptide analogs interfering with the immune system in the same way as native peptides of pneumococcal proteins.

Thorough understanding of the interaction of the immune system with the protein fragments is being currently elucidated. The possibilities for immune modulation using peptide antigens seem very broad and will depend on a more comprehensive knowledge of the underlying molecular mechanisms influencing this type of recognition. Innovative approaches in synthesis, structural analysis, conjugation, delivery, and immuno-monitoring will greatly increase the value of peptides in the design of novel pneumococcal vaccines.

## Figures and Tables

**Table 1 vaccines-07-00009-t001:** Summary of pneumococcal proteins that have been evaluated or are currently under evaluation as components of protein-subunit vaccines in humans.

Vaccine	Assignee	Current Status	References
PhtD monovalent	GSK	Phase 2 completed	Leroux-Roels et al., 2014 [41] Seiberling et al., 2012 [42]
PhtD monovalent	Sanofi Pasteur	Phase 1 completed	Leroux-Roels et al., 2014 [41]
PhtD+dPly	GSK	Phase 2 completed	Seiberling et al., 2012 [42]
Protein D+PhtD+dPly trivalent	GSK	Phase 1 completed	Berglund et al., 2014 [43]
PhtD+dPly+PCV8	GSK	Phase 1 completed	Seiberling et al., 2012 [42]
PhtD+dPly+PCV10	GSK	Phase 2 completed	Seiberling et al., 2012 [42]
dPly, PhtD+dPly+PCV10	GSK	Phase 1 completed	Seiberling et al., 2012 [42]
PspA	Sanofi Pasteur	Phase 1 completed	Frey et al., 2013 [44]
PlyD1 monovalent	Sanofi Pasteur	Phase 1 completed	Bologa et al., 2012 [45] Kamtchoua et al., 2013 [46]
PspA+PsaA	Sanofi-Pasteur/CDC	Phase 1 completed	Briles et al., 2000 [47,48] Nabors et al., 2000 [49]
PcpA	Sanofi Pasteur	Phase 1 completed	Bologa et al., 2012 [45]
PcpA+PhtD	Sanofi Pasteur	Phase 1 completed	Khan et al., 2012 [50] Bologa et al., 2012 [45]
PhtD+PcpA+PlyD1	Sanofi Pasteur	Phase 1 completed	Bologa et al., 2012 [45]
PcsB, StkP, PsaA (IC-47)	Intercell AG/Novartis	Phase 1 completed	Kamtchoua et al., 2013 [46]

**Table 2 vaccines-07-00009-t002:** The most studied proteins regarding the identification of the antigenic regions they contain.

Protein Name	Function	Virulence Factor	Location	3-D structure of the Full Length Proteins
Pneumococcal surface protein A (PspA)	Cellular Metabolism and Immune Evasion	Choline-Binding Proteins (CBPs)	Bound to the cell wall via PCho moiety	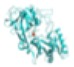
Pneumococcal Choline-Binding Protein A (PcpA)	Protection Against Lung Infection and Sepsis	Choline-Binding Proteins (CBPs)	Bound to the cell wall via PCho moiety	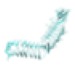
Secreted 45-KDa Protein, Usp45-Hydrolase (PcsB)	Virulence	Non-Classical Surface-Exposed Proteins	Membrane	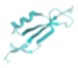
Serine/Threonine Protein Kinase (StkP)	Cellular Metabolism and Fitness	Non-Classical Surface-Exposed Proteins	Membrane	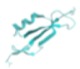
Peptide Permease Enzyme, Manganese ABC Transporter (PsaA)	Immune Evasion	Lipoproteins	Surface of the cell wall	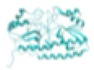
Pneumolysin D (PlyD)	Cytolytic Toxin, Adherence, Immune Evasion, Invasion, Dissemination and Complement Activation	Non-Classical Surface-Exposed Proteins	Cytoplasmic toxin	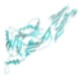
Pneumolysin toxoid (dPly)	Toxoid	Non-Classical Surface-Exposed Proteins	Toxoid	N.A. *
Choline-binding protein D (CbpD)	Colonization	Choline-Binding Proteins (CBPs)	Bound to the cell wall via PCho moiety	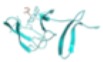
Histidine triad protein D (PhtD)	Adherence and Immune Evasion	Lipoproteins	Surface of the cell wall	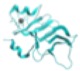
Histidine triad protein E (PhtE)	Adherence and Immune Evasion	Lipoproteins	Surface of the cell wall	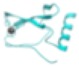
Histidine triad protein A (PhtA)	Adherence and Immune Evasion	Lipoproteins	Surface of the cell wall	N.A. *
Histidine triad protein B (PhtB)	Adherence and Immune Evasion	Lipoproteins	Surface of the cell wall	N.A. *
Plasmin and Fibronectin-Binding Protein A (PfbB)	Adherence, Immune Evasion and Antiphagocytosis	LPxTG—Proteins	Anchored to well wall by the enzyme sortase A	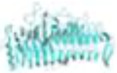
Zinc metalloproteinase B (ZmpB)	Immune Evasion and Colonization	LPxTG—Proteins	Anchored to well wall by the enzyme sortase A	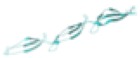

* The 3-D structure of the corresponding protein has not been defined yet.

**Table 3 vaccines-07-00009-t003:** Particular vaccine adjuvants or delivery systems used in peptide-based pneumococcal vaccines in pre-clinical studies so far.

Protein	Fragment	Adjuvant	Reference
Pneumococcal surface proteins: CbpD, PhtD, PhtE, ZmpB	CbpD_pep4, PhtD_pep19, PhtE_pep40, ZmpB_pep125 and	Freund’s adjuvant	Papastamatiou et al., 2018 [96]
PspA	α1α2 fragment	OMVs (Salmonella Outer Membrane Vesicles)	Kuipers et al., 2017 [87]
PspA, PdT	PspA-PdT fusion protein	BCG (Bacillus Calmette-Guérin)	Goulart et al., 2017 [103]
PspA, PotD	PspA-PotD fusion protein	Al(OH)_3_	Converso et al., 2017 [104]
PspA	linear peptides from the PRR or the NPB	SVLPs (synthetic Virus-Like Particles)	Tamborrini et al., 2015 [90]
PspA, Ply	N-terminal fragments of PspA, N-terminal & C-terminal fragments of Ply, fused to an Hbp-carrier	OMVs (Salmonella Outer Membrane Vesicles)	Kuipers et al., 2015 [86]
Ply, CbpA, PspA	combination of full-length or peptide regions of Ply, CbpA, or PspA	Alum	Chen et al., 2015 [105]
PspA,	PspA epitopes from the N-terminal region	Aluminum hydroxide	Vadesilho et al., 2014 [84]
SP_2108 and SP_0148 lipoproteins	2108- 1912 (fusion construct of SP_2108 and SP_0148)	Cholera Toxin (CT)Aluminum hydroxide	Moffitt et al., 2014 [113]
PhtD	full length PhtD and truncated derivatives (PhtD, PhtD C1, PhtD N2 etc.)	E. coli labile toxin B subunit (LTB)	Plumptre et al., 2013 [95]
Protein encoded by the spr1875 of R6 strain genome	R4-GST fusion protein (glutathione S-transferase) (R4: 161 amino acid-long fragment)	Freund’s adjuvant	Cardaci et al., 2012 [100]
ZmpB, Ply, Dnal	rZmpB, rPly, rDnaJ, rZmpB + rPly, rZmpB + rDnaJ, rPly + rDnaJ, rZmpB + rPly + rDnaJ,	Freund’s adjuvant	Gong et al., 2011 [98]
PspA	N-terminal regions of clades 1,3 and 4 +PRR, N-terminal-PRR, N-terminal+ first block of PRR	Al(OH)_3_	Darrieux et al, 2007 [83]
PspA	PR regions and NPB regions (PR+NPB _C, PR+NPB _N, PR+NPB, PR-NPB, or NPB)	Alum	Daniels et al., 2010 [82]
PspA	recombinant N-terminal His- tagged PspA protein	cholera toxin B subunit (CTB)	King et al., 2009 [89]
PspA	Recombinant PspAs	TLR-agonists: Pam3 Ultra Pure Escherichia coli K12 LPS, CSK4, Poly(I:C) and	Oma et al., 2009 [112]
Several proteins (i.e., PspA, PscB, StkP, PhtD etc)	N-terminal two thirds of PcsB, C-terminal half of StkP	Alum	Giefing et al., 2008 [67]
PspA	Six overlapping recombinant fragments of family 2, clade 3 PspA/EF3296	AlOH_3_	Roche et al., 2003 [125]
PhpA	Recombinant PhpA-79	AlPO_4_	Zhang et al., 2001 [92]
PsaA	peptide 43 (palmitoyl residue at the N-terminal end), 46 (same peptide without any ligands, nonlipidated), and PsaA	Aluminum hydroxide	Srivastava et al., 2000 [99]

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
