# Peer review of "Novel Protein-Based Pneumococcal Vaccines: Assessing the Use of Distinct Protein Fragments Instead of Full-Length Proteins as Vaccine Antigens"

_vaccines, 2019, doi:10.3390/vaccines7010009_

Round 1

Reviewer 1 Report

In this review, Lagousi et al. describe approved and experimental pneumococcal vaccines. They begin with describing the limitations of the current vaccine, followed by describing the use of protein and peptide-based vaccines.

General comments:

Overall the wording in manuscript, especially the first half, reads awkwardly. Rewriting sentences for clarity, including removing filler words/phrases that are the first word or linker words throughout the document would help. These include: besides nevertheless, yet, firstly, in a step forward, etc.

Specific comments:

Line 31: Bacterial names should be italicized

Line 31: The comma after “peneumoniae” should be removed and “while” should be changed to “and” for sentence clarity.

Line 53: more should be changed to most (if it is second on the list)

Lines 76-77: Please remove “as held until then” for sentence clarity.

Line 78, 80,113, 117, 121: Please provide the appropriate references

Lines 104-107: Do any of the candidate vaccines listed in Table 1 contain any of the 26 proteins that are constitutively expressed/low sequence variation in reference 44?

Line 109: It is usually the protein’s 3D structure…

Line 143:” ..are capable to induce..” should read “are capable of inducing”

Line 163: remove the word “to”

Table 2 is fuzzy and needs a title for the abbreviated protein names (or put them in the same column). It is also unclear what the structures on the right are showing, the brackets, and the N.A.s are referring to.

Author Response

In this review, Lagousi et al. describe approved and experimental pneumococcal vaccines. They begin with describing the limitations of the current vaccine, followed by describing the use of protein and peptide-based vaccines.

General comments:

Overall the wording in manuscript, especially the first half, reads awkwardly. Rewriting sentences for clarity, including removing filler words/phrases that are the first word or linker words throughout the document would help. These include: besides nevertheless, yet, firstly, in a step forward, etc.

Response: Thank you for your comment. Most filler words/phrases that are the first word or linker words throughout the document have been removed. 

Specific comments:

Line 31: Bacterial names should be italicized. 

Response: Thank you for your comment. Bacterial name has been changed to italics (line 32).

Line 31: The comma after “peneumoniae” should be removed and “while” should be changed to “and” for sentence clarity. 

Response: Thank you for your comment. Manuscript has been rephrased  (line 32).

Line 53: more should be changed to most (if it is second on the list). 

Response: Thank you for your comment. More has been replaced by most (line 54).

Lines 76-77: Please remove “as held until then” for sentence clarity.

Response: Thank you for your comment. The above phrase has been removed (lines 99-100).

Line 78, 80,113, 117, 121: Please provide the appropriate references

Response: Thank you for your comment. The corresponding references have been added. Please refer to lines 100, 101, 137, 141 and 145 respectively. 

Lines 104-107: Do any of the candidate vaccines listed in Table 1 contain any of the 26 proteins that are constitutively expressed/low sequence variation in reference 44? 

Response: Thank you for your comment. “Remarkably, only 3 of the above 26 widely expressed proteins (namely pneumococcal surface antigen A (PsaA), serine threonine kinaseprotein(StkP) and pneumolysin), have been so far included as candidate antigens in the vaccines that are currently under evaluation in clinical studies listed in Table 1.” (lines 130-133).

Line 109: It is usually the protein’s 3D structure…

Response: Thank you for your comment. Manuscript has been rephrased (lines135-136).

Line 143:”are capable to induce..” should read “are capable of inducing”

Response: Thank you for your comment. Manuscript has been rephrased (line 204).

Line 163: remove the word “to”.

Response: Thank you for your comment. “To” has been removed (line 225).  

Table 2 is fuzzy and needs a title for the abbreviated protein names (or put them in the same column). It is also unclear what the structures on the right are showing, the brackets, and the N.A.s are referring to.

Response: Thank you for your comment. Table 2 has been modified to be more comprehensible (lines 241-242).

Reviewer 2 Report

All currently available Streptococcus pneumoniae (Spn) vaccines have limitations due to their capsular serotype composition. Both the 23-valent Spn polysaccharide vaccine (PPV) and 7, 10, or 13-valent Spn conjugate vaccines (PCV-7, 10, -13) are serotype-based vaccines and therefore they elicit only serotype-specific immunity. This review is well written, discusses the progress to date on development of Spn protein vaccine candidates, with special focus on the combination of several antigenic regions or peptides that may impair different stages of pneumococcal disease, offering even wider serotype coverage and more efficient protection. Furthermore, particulate mention is given on the use of newer delivery system technologies including conjugation to TLRs and reformulation into nanoparticles to enhance the poor immunogenicity of such antigens. All makes this review more comprehensive and updated.

Several comments as below;

1.     Many abbreviation/Jargon loaded in this manuscript without being defined at all or upon first mention. For example, IPD: invasive pneumococcal disease, PspA: pneumococcal surface protein A (PspA) and C-reactive protein (CRP). Technical fields are loaded with abbreviations and acronyms whose meanings experts take for granted. In a manuscript, however, abbreviation overuse can instead reduce readability, forcing a non-specialist reader to pause and refer back to the original definition; it is worse when the definition doesn’t even exist. If you have decided to use particular abbreviations, note that it typically should be defined upon first mention in both the abstract and the main text and then applied consistently throughout the remainder of the respective sections.

2.     NP colonization has been one of the key aspects in evaluating the PCVs because elimination of colonization provides a herd immunity effect. Most cost-effectiveness calculations take into account the impact of herd protection, and in a varying degree the replacement of disease. Author may need to update and compare capsular polysaccharide versus protein-based vaccine effects on NP colonization.

3.     In addition, historically vaccines were developed using methods that reduced or inactivated a pathogen’s virulence; such as attenuation or chemical treatment or preparations of whole-cell crude extracts. For example, both whole-cell and acellular purified protein-based pertussis vaccines provide good protection, however, whole-cell pertussis vaccine are more efficacious and produce longer lasting immunity than acellular vaccine. Based on the observations regarding whole cell pertussis vaccines, a pneumococcal WCV may be the best strategy for developing countries. Any update?

4.     Any discussion and update for pneumococcal protein-based vaccines to prevent acute otitis media? AOM is the best type of pneumococcal infection to be used in assessing efficacy of WCVs and PPVs, because the etiology of AOM can be proven using tympanocentesis and because of higher costs and longer time to accumulate a sufficient number of subjects to prove efficacy of invasive pneumococcal disease with concurrent PCV administration.

Author Response

All currently available Streptococcus pneumoniae (Spn) vaccines have limitations due to their capsular serotype composition. Both the 23-valent Spn polysaccharide vaccine (PPV) and 7, 10, or 13-valent Spn conjugate vaccines (PCV-7, 10, -13) are serotype-based vaccines and therefore they elicit only serotype-specific immunity. This review is well written, discusses the progress to date on development of Spn protein vaccine candidates, with special focus on the combination of several antigenic regions or peptides that may impair different stages of pneumococcal disease, offering even wider serotype coverage and more efficient protection. Furthermore, particulate mention is given on the use of newer delivery system technologies including conjugation to TLRs and reformulation into nanoparticles to enhance the poor immunogenicity of such antigens. All makes this review more comprehensive and updated.

Several comments as below:

1. Many abbreviation/Jargon loaded in this manuscript without being defined at all or upon first mention. For example, IPD: invasive pneumococcal disease, PspA: pneumococcal surface protein A (PspA) and C-reactive protein (CRP). Technical fields are loaded with abbreviations and acronyms whose meanings experts take for granted. In a manuscript, however, abbreviation overuse can instead reduce readability, forcing a non-specialist reader to pause and refer back to the original definition; it is worse when the definition doesn’t even exist. If you have decided to use particular abbreviations, note that it typically should be defined upon first mention in both the abstract and the main text and then applied consistently throughout the remainder of the respective sections.

Response: Thank you for your comment.  We have defined upon first mention all the particular abbreviations throughout the abstract as well as the main text and then used them consistently in the following sections. 

2. NP colonization has been one of the key aspects in evaluating the PCVs because elimination of colonization provides a herd immunity effect. Most cost-effectiveness calculations take into account the impact of herd protection, and in a varying degree the replacement of disease. Author may need to update and compare capsular polysaccharide versus protein-based vaccine effects on NP colonization.

Response: Thank you for your comment. Please refer to lines 150-174 of the revised manuscript for further clarification of the different effects of polysaccharide-based and protein-based pneumococcal vaccines on nasopharyngeal (NP) colonization.

"Another issue worth mentioning is the different effects of polysaccharide-based and protein-based pneumococcal vaccines on NP colonization. NP colonization is an essential precursor to pneumococcal disease pathogenesis. In the NP, the expression of various pneumococcal virulence factors differs between transparent and opaque phase, facilitating the establishment of commensal carriage. Therefore, antibodies against capsular polysaccharides fail to protect against already established NP colonization, while antibodies against proteins may hamper initiation of colonization. Besides, antibodies against capsular polysaccharides prevent carriage of a specific serotype leading to a significant herd immunity effect; however, complete elimination of carriage in the NP and subsequent replacement of vaccine serotypes by non-vaccine serotypes has been an important issue. In contrast, protein-based pneumococcal vaccines, that are serotype independent, have been associated with intermittent priming and boosting to all pneumococcal serotypes through natural exposure. This effect has been suggested to retain pneumococcal carriage density below disease threshold, instead of the total elimination induced by polysaccharide-based vaccines. Thus, such vaccines may induce more robust memory responses, and at the same time, prevent other pathogens to emerge in the nasopharynx. A recent study among healthy infants showed that NP mucosal antibody levels to pneumococcal histidine triad D (PhtD), pneumococcal-choline binding protein A (PcpA) and pneumolysin detoxified derivative (PlyD1), induced through repeated colonization, correlated with protection against acute otitis media (AOM) but not protection against NP colonization. Additionally, in a mouse influenzae/pneumococcal co-infection model it was shown that vaccination with PhtD prevented pneumococcal density from reaching a pathogenic threshold during a viral upper tract respiratory infection (URTI) without elimination of the pathogen from the NP. However, things may differ in developing countries, where persistently high antigenic exposure attributed to the high pneumococcal burden might lead to immune hyporesponsiveness, resulting in low vaccine efficacy, since antigen specific CD4 T-cells can lose their ability to help B cells differentiate into antibody secreting cells. "

3. In addition, historically vaccines were developed using methods that reduced or inactivated a pathogen’s virulence; such as attenuation or chemical treatment or preparations of whole-cell crude extracts. For example, both whole-cell and acellular purified protein-based pertussis vaccines provide good protection, however, whole-cell pertussis vaccine are more efficacious and produce longer lasting immunity than acellular vaccine. Based on the observations regarding whole cell pertussis vaccines, a pneumococcal WCV may be the best strategy for developing countries. Any update?

Response: Thank you for your comment. Please refer to lines 63-85 of the revised manuscript for the additional information regarding whole cell pneumococcal vaccines.

"Another approach employs the use of next-generation whole-cell (killed) pneumococcal vaccines (WCPVs). So far, several methods including attenuation, chemical treatment or preparations of whole-cell crude extracts have been used to reduce or inactivate a pathogen’s virulence for the development of vaccines. For example, both whole-cell and acellular purified protein-based pertussis vaccines provide efficient protection; however, whole-cell pertussis vaccine are more efficacious and produce longer lasting immunity than acellular vaccine. Based on such observations, WCPVs may be safe, provide multiple common surface proteins in native configurations and induce an immune response that is protective across different serotypes.

A WCPV may be the most suitable strategy for developing countries, as it is inexpensive and easy to construct and administer. 

One such vaccine containing a non-capsulated pneumococcal strain has shown protection against nasopharyngeal (NP) colonization and sepsis in immunized mice. Protection against NP colonization was shown to be CD4+ Interleukin 17A (IL-17A)-dependent and non-capsular antibody mediated.

The WCPV was used in two clinical trials to study safety, dose tolerance and immunogenicity.  In healthy US adults (NCT01537185) WCPV was shown to be safe and well tolerated; significant IgG responses to pneumococcal antigens were elicited, including pneumococcal surface protein A (PspA) and pneumolysin, and functional antibody responses were detected in a pneumolysin toxin neutralizing assay. Significant increases in T-cell cytokine responses, including IL-17A, were measured among subjects receiving the highest dose of WCPV. Passive transfer of human immune sera protected mice against fatal sepsis when challenged intravenously with a virulent strain of pneumococci. further studies are required to evaluate the safety, dose tolerance and immunogenicity of such vaccines."

4. Any discussion and update for pneumococcal protein-based vaccines to prevent acute otitis media? AOM is the best type of pneumococcal infection to be used in assessing efficacy of WCVs and PPVs, because the etiology of AOM can be proven using tympanocentesis and because of higher costs and longer time to accumulate a sufficient number of subjects to prove efficacy of invasive pneumococcal disease with concurrent PCV administration.

Response: Thank you for your comment. Please refer to lines 176-193 of the revised manuscript regarding the update information for protein-based pneumococcal vaccines against AOM.

"AOM is the most suitable type of pneumococcal infection to assess the efficacy of pneumococcal vaccines due to its easy diagnosis using tympanocentesis and lower costs and shorter time to accumulate a sufficient number of subjects compared to IPD with concurrent PCV administration. Mucosal immunity plays a critical role in control of pneumococcus locally invasive infections including AOM. Capsular polysaccharide vaccines are less effective in protection against AOM than against IPD. As mentioned above, NP mucosal antibody levels to PhtD, PcpA and PlyD1, induced through repeated colonization, correlated with protection against AOM. Similarly, higher levels of mucosal antibodies against pneumococcal protein antigens(PhtD, Pneumococcal histidine triad protein E (PhtE), the endo-β-N-acetylglucosaminidase (LytB), PcpA, PspA and Ply) induced through natural exposure in healthy children were associated with reduced risk of AOM. 

Currently, a clinical efficacy study is underway in an otitis-prone Native American population (NCT01545375). Nevertheless, for a vaccine to be efficient against AOM, special attention should be given to studies in the infection-prone child who has an immune response that is generally deficient compared to non-otitis prone children leading to susceptibility to recurrent AOM.

Therefore, while otitis-prone population may be attractive to assess the efficacy of protein-based pneumococcal vaccines against AOM, it should be realized that the response to such vaccines in otitis-prone populations may not be of the same magnitude as the general population."